# Embodied Instruction Following in Unknown Environments

## Abstract

Enabling embodied agents to complete complex human instructions from natural language is crucial to autonomous systems in household services. Conventional methods can only accomplish human instructions in the known environment where all interactive objects are provided to the embodied agent, and directly deploying the existing approaches for the unknown environment usually generates infeasible plans that manipulate non-existing objects. On the contrary, we propose an embodied instruction following (EIF) method for complex tasks in the unknown environment, where the agent efficiently explores the unknown environment to generate feasible plans with existing objects to accomplish abstract instructions. Specifically, we build a hierarchical embodied instruction following framework including the high-level task planner and the low-level exploration controller with multimodal large language models. We then construct a semantic representation map of the scene with dynamic region attention to demonstrate the known visual clues, where the goal of task planning and scene exploration is aligned for human instruction. For the task planner, we generate the feasible step-by-step plans for human goal accomplishment according to the task completion process and the known visual clues. For the exploration controller, the optimal navigation or object interaction policy is predicted based on the generated stepwise plans and the known visual clues. The experimental results demonstrate that our method can achieve 45.09% success rate in 204 complex human instructions such as making breakfast and tidying rooms in large house-level scenes.

## 1 Introduction

Building intelligent autonomous systems (Huang et al., 2023; Mu et al., 2024; Brohan et al., 2022; Ahn et al., 2022) to complete household tasks such as making breakfast and tidying rooms is highly demanded to reduce the laborer cost in our daily life. The agent is required to understand the visual clues of the surrounding scene and the language instructions, and feasible action plans are then generated for object interaction with the goal of high success rate and low action cost to accomplish human demands.

To achieve this, end-to-end methods (Pashevich et al., 2021; Zhang & Chai, 2021; Van-Quang Nguyen, 2020) directly generate the low-level actions from raw image input and natural language with the supervision of expert trajectories. To reduce the learning difficulties in the complex task, modular methods (Ding et al., 2023; Inoue & Ohashi, 2022; Murray & Cakmak, 2022; Liu et al., 2022), sequentially learn the instruction comprehension, state perception, spatial memory construction, high-level planning and low-level control to complete human goals. Since embodied agents are expected to complete more diverse and complex instructions, large language models (LLMs) are widely employed in EIF (Lu et al., 2023; Wu et al., 2023; Gordon et al., 2018; Misra et al., 2017; Shah et al., 2023) due to their strong reasoning power and high generalization ability. However, existing methods can only generate plans in known environments where categories of all interactable objects in the scene are given to LLMs. Since the agent does not know the objects in the unknown environment, the generated plans are usually infeasible because of interacting with non-existing objects. Figure 1 (a) demonstrates an example of existing methods, where the agent is unaware that no bottles exist in the unknown environment. Interacting with the non-existent bottles based on the infeasible plan fails to accomplish the human goals of water serving.

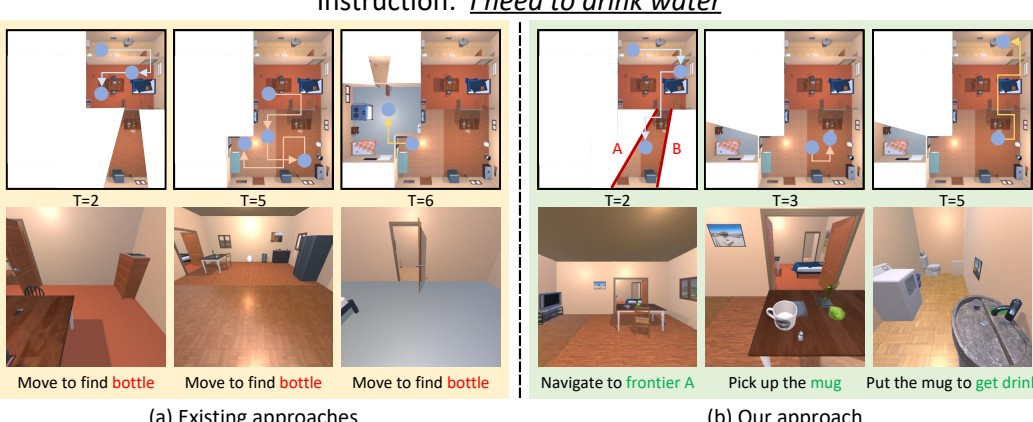

Figure 1: Comparison between conventional EIF methods and our approach in unknown environments. Existing methods fail to complete the instruction even with long exploration cost, while our method efficiently achieves the goal with efficient navigation and object interaction.

In realistic deployment scenarios, household agents usually work in unknown environments without stored scene maps. Building scene maps in advance cannot accurately represent the scene, where object properties such as location and existence change frequently due to human activity in daily life. For example, the mug may be on the dining table and the coffee table respectively when humans are having dinner and watching TV. Meanwhile, potatoes might have been consumed and tomatoes are then purchased for the next breakfast. Therefore, failing to generate feasible plans in unknown environments strictly limits the practicality of the embodied agents. The agent working in realistic deployment scenarios is required to build real-time scene maps, where feasible plans are generated with minimal exploration cost.

In this paper, we propose an EIF method for complex tasks in the unknown environment. Different from conventional methods that assumes knowing interactable objects in advance, our method navigates the unknown environment to efficiently discover objects that are relevant to the complex human requirements. Therefore, the embodied agent can generate feasible task plans in realistic indoor scenes where the locations and existence of objects are frequently changing. Figure 1 (b) also demonstrates the same example of water serving implemented by our method, and our agent efficiently discovers the mug and uses it as the receptacle of water because no bottles exist in the scene. We first construct a hierarchical EIF framework including the high-level task planner and the low-level exploration controller with multi-modal LLMs, which are finetuned by the large-scale generated trajectories of the complex EIF tasks. We then design a scene-level semantic representation map to depict the visual clues in the known area, through which the goals of the task planner and the exploration controller can be aligned to feasibly complete human instructions.

More specifically, the goal of the task planner is to generate feasible plans for human instruction including navigation and manipulation in natural language. The task planner predicts the next step based on the semantic representation map and the task completion process. The exploration controller aims at discovering task-related objects with low action cost, which selects the optimal navigation policy from all navigable borders or object interaction policy according to the semantic representation map and the generated step-wise plans. For the scene-level semantic feature map, we project the CLIP features of collected RGB images during exploration to the top-down map with dynamic region attention, which preserves the task-relevant visual information in the map without redundancy. The experimental results in ProcTHOR (Deitke et al., 2022) simulation environment show that our method can achieve 45.09% success rate in 204 complex human instructions in large house-level scenes.

## 2 RELATED WORKS

**Embodied Instruction Following:** The EIF task requires the robot to follow human instructions represented by natural language in the interactive environment. A key challenge for the EIF task

is generating interaction goals and actions grounded in the deployment environment according to the instructions. Prior works (e.g., LACMA (Yang et al., 2023), E.T. (Pashevich et al., 2021), M-TRACK (Song et al., 2022)) have explored end-to-end transformer architecture to generate grounded low-level interaction actions based on the current environment perception, modular approaches (e.g., HLSM (Blukis et al., 2022), FILM (Min et al., 2021), LLM-Planner (Song et al., 2023)) propose enhancing the generalization of unseen scenes with hierarchical planners. However, prior arts have focused on single-room environments, which are designed for known environments where visual clues of the whole scene can be easily acquired by looking around. The low scalability of the scene scale limits their ability to discover required visual clues in unknown environments for feasible action generation.

**Scene Representation for Visual-language Navigation:** Visual-language navigation requires agents to explore unknown environments to locate target objects and follow natural language instructions. The primary challenge lies in efficiently representing expansive unknown scenes for generating navigation policies. Existing scene representations consist of three categories: 2D semantic maps, 3D geometric maps and scene graphs. Early works (Batra et al., 2020; Anderson et al., 2018) constructed the 2D semantic maps by projecting visual clues in the top-down view, which are leveraged for navigation frontier selection for target finding. PONI (Ramakrishnan et al., 2022) proposed a scoring network for all potential frontiers of unseen regions 2D semantic maps, and L3MVN (Yu et al., 2023) determined the semantic relevance of the objects around each frontier to the target by BERT (Devlin et al., 2018). To embed the geometric information, 3D geometric maps are investigated by fusing the structure and semantic information. LERF (Kerr et al., 2023) and ConceptFusion (Jatavallabhula et al., 2023) integrated fine-grained alignment of semantic features with 3D maps in SLAM, multi-view fusion, and NeRF (Mildenhall et al., 2021) for multiple downstream tasks. To reduce the storage overhead, scene graphs (Gu et al., 2023; Hughes et al., 2022) are proposed to represent objects or concepts as nodes and spatial relations as edges to represent the scene topology efficiently. SayPlan (Rana et al., 2023) enabled agents to focus on task-relevant nodes by integrating subgraph folding and replanning mechanisms. Inspired by the above approaches, we construct semantic feature maps to empower embodied agents to explore unknown environments, where task-relevant information can be acquired for action generation with low exploration cost.

## 3 PROBLEM STATEMENT

Given the human instruction $I$ in natural language, the robot should generate a sequence of action primitives including (`PickUp`, `Place`, `Open`, `Close`, `ToggleOn`, `ToggleOff`, `Slice`) to complete the instruction. The agent can only acquire the scene information for instruction following via an RGB-D camera mounted on the agent, through which the agents build a semantic map $S$ to generate the feasible interaction. In realistic deployment, the embodied agent usually work in unknown environments, where the location and existence of objects in the house-level scene are not known. Therefore, we add an additional action primitive (`Navigation`) to enable the agent to explore the scene for visual information collection.

The agent consists of a high-level planner that reasons step-by-step plans $P = \{p_i\}_{i=1}^T$ from human instructions and a low-level controller that predicts the specific actions $A = \{a_j^i\}_{j=1}^{\tau_i}$ for each step for scene navigation or object interaction. $T$ means the number of steps to achieve the human goal, and $\tau_i$ is the number of special actions to achieve the $i_{th}$ step in the high-level plan. The high-level planner is represented by natural language (e.g. Step 2. Heat the potato) given the human instruction (e.g. Can you make breakfast for me?), and the low-level controller transfers the step-by-step plans into executable actions with action primitives, location and target objects (e.g. `Place`, potato, (10, 8) or `Navigate`, frontier, (2, 3)). Finally, the agent only manipulates the existing relevant objects to achieve human goals.

## 4 APPROACH

In this section, we first introduce the overall pipeline of our EIF method designed for unknown environments, and then we describe the details of the high-level planner and the low-level controller. Moreover, we elaborate the construction of the online semantic feature maps that ground the planner

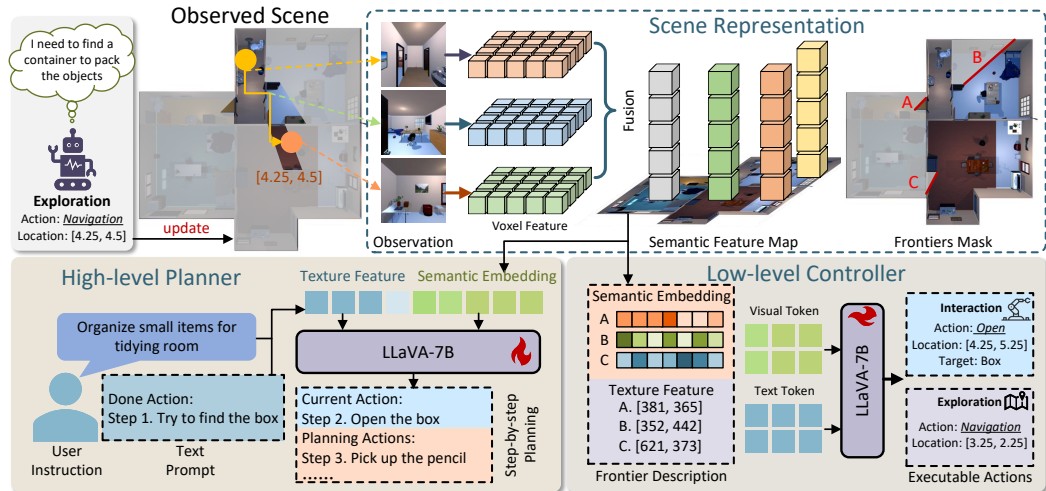

Figure 2: Overview of our approach. The scene feature map is constructed based on real-time RGB-D images, which is leveraged as visual clues for the high-level planner and the low-level controller. The planner generates the step-wise plans, which are leveraged to predict the specific actions in the controller. The optimal border between unknown and known regions is selected for scene exploration, and the scene feature map is updated with the visual clues seen in during the exploration.

and the controller to the physical scene. Finally, we demonstrate the model training and the inference of our framework in practical deployment.

## 4.1 OVERALL PIPELINE

In realistic deployment scenarios of household robots, the physical world is usually unknown for the agent because the existence and locations frequently change due to human activity. Therefore, the agents are required to construct the online scene feature map according to the real-time visual perception during the robot navigation, through which the agent generates feasible step-by-step plans to achieve the human goal and the efficient exploration trajectories for the unknown scene including navigation and object interaction to complete each step in the plan. Figure 2 demonstrates the overall pipeline of our agent. The scene feature map represents the visual clues of the scene in the top-down view based on the collected RGB-D images during exploration, where the pre-trained features of regions with higher relevance to the instruction are assigned with higher importance for feature map construction. The high-level planner generates the plans for the next step with natural language based on the task completion process and the semantic feature map, and the low-level controller predicts the templated action primitives, location and target objects for executable navigation or manipulation based on the scene feature map and the plan for the next step.

## 4.2 HIERARCHICAL EMBODIED AGENTS FOR EIF IN UNKNOWN ENVIRONMENTS

We decompose EIF in unknown environments into two sub-tasks including the high-level planning and the low-level exploration. The generated high-level plans are leveraged as guidance for the agent to select the most relevant regions for exploration, and the predicted low-level actions update the semantic feature maps to provide visual clues for feasible plan generation. Both the planner and the explorer are implemented by a finetuned LLaVA model.

**High-level planner:** The planner generates the plan for the next step in natural language, which considers the textual information including the human instruction and the completed steps and the visual clues represented by the semantic feature maps. The forward pass of the high-level planner $HP$ can be represented as follows:

$$p_i = HP(I, \{p_k\}_{k=1}^{i-1}; S_{i-1}) \tag{1}$$

where $S_i$ means the semantic feature maps updated in the $i_{th}$ step and we leverage a LLaVA model whose visual encoder is the ViT-L/14 architecture for the high-level planner.

**Low-level controller:**    The low-level controller predicts the specific actions including the action primitives, locations, and target objects according to the generated high-level plans and the semantic feature maps, which explores the unknown scene and completes the step-wise plan. The forward pass of the low-level controller $LC$ can be represented as follows:

$$\{a_j^i, l_j^i, o_j^i\} = LC(p_i, \{f_m^i\}_m; \{s_m^i\}_m) \tag{2}$$

where $l_j^i$ and $o_j^i$ are the predicted location and target objects for the $j_{th}$ actions in the $i_{th}$ step of the high-level plan. Meanwhile, $f_m^i$ means the textual features of the $m_{th}$ segment of the frontier between known and unknown regions for $S_i$, where $m$ represents the number of frontier segments in the entire $S_i$. The textual features are demonstrated by the coordinate of the middle point for the frontier segment. $s_m^i$ denotes the semantic features of the $m_{th}$ frontier segments, which is demonstrated by the semantic feature map patches containing the corresponding frontiers. The low-level controller not only explores the unknown scene with navigation and object interaction but also completes the step-wise plans by manipulating the target object (e.g. pick up the tomato). For action primitives except for `navigate`, the predicted actions are implemented on the target objects. For `navigate`, the robot just moves to the predicted locations without object interaction.

### 4.3    Online Semantic Feature Maps

The high-level planner and the low-level controller should be aligned so that they can generate feasible plans and exploratory actions to achieve human instructions in the unknown environment. The semantic feature maps can be leveraged for alignment since they provide visual clues of the scene for both the high-level planner and the low-level controller. In realistic deployment scenarios of household robots, the existence and locations frequently change due to human activity. Therefore, we propose an online semantic feature map that is dynamically updated during the exploration of the unknown scene for each human instruction.

Semantic feature maps represent the visual cues from image observations in top-down view. Compared with simple semantic maps which store the object categories of pixels, our semantic feature maps can represent implicit relationships between objects in the scene, which provides crucial information for effective exploration policy generation. For EIF in unknown environments, the visual information collected in the $i_{th}$ timestep contains the RGB image $C_i$ and the depth image $D_i$. To enable the semantic feature maps to acquire high generalization ability in diverse human instructions, we leverage CLIP to extract the pixel-wise visual features $\mathbf{f}_{xy}^i$ at time $i$ for the pixel in $x_{th}$ row and $y_{th}$ column of $C_i$ by fusing the feature of the entire image and that of the instance mask containing the corresponding pixel. The visual features contribute to the projected location in the scene feature map in the top-down view, which can be depicted as follows:

$$\mathbf{F}_{uv}^i = \sum_{x,y} \mathbf{f}_{xy}^i \cdot \mathbb{I}(\mathcal{P}((x,y), D_i) \in \mathcal{S}(u,v)) \tag{3}$$

where $\mathbf{F}_{uv}^i$ means the contribution to the element in the $u_{th}$ row and $v_{th}$ column of the semantic feature map from the visual information collected in time $i$, and $\mathcal{P}((x,y), D_i)$ demonstrates the projected coordinates in the top-down view for of the pixel $(x,y)$ based on the depth image $D_i$. $\mathcal{S}(u,v)$ means the pixel in the $u_{th}$ row and $v_{th}$ column in the semantic feature map, and the indicator function $\mathbb{I}(\cdot)$ equals one for true and zero otherwise.

The semantic feature map is updated at each time step during the exploration process, where the agent observes new visual information for recording. Since the house for embodied instruction following in realistic world is usually very large, regarding all images with equal importance in semantic feature map construction leads to significant information redundancy. Meanwhile, different visual clues usually make various contribution to the given human instruction. Therefore, we should assign large importance to relevant visual clues when updating the semantic feature maps, so that sufficient visual information can be represented without redundancy for high-level planning and low-level exploration. The task relevance can be acquired as follows. The high-level planner is also required to generate the demanded objects $\{O_k\}_k$ for the predicted corresponding step-wise plan, which are leveraged to construct three prompts including (a) the image contains $\{O_k\}_k$, (b) the image does not contain $\{O_k\}_k$ and (c) the image contains nothing. We then leverage a pre-trained LongCLIP (Zhang et al., 2024) to predict the similarity score between the image and all prompts.

Finally, the online semantic feature map is updated with dynamic region attention:

$$\mathbf{S}_{uv}^{i} = (1 - w_i)\mathbf{S}_{uv}^{i-1} + w_i\mathbf{F}_{uv}^{i}, \quad w_i = c_i / \frac{1}{i}\sum_{k=1}^{i} c_k \qquad (4)$$

where $\mathbf{S}_{uv}^{i}$ means the features in the $i_{th}$ row and $j_{th}$ column of the semantic feature maps at time $i$. The normalized weight $w_i$ represents the importance of the current semantic features compared with known visual clues, where $c_k$ is the original similarity score between the image and the prompt in the $k_{th}$ time step. The online semantic feature maps contain rich visual information, and the most relevant regions can be explored via navigating the optimal border and interacting with related objects to achieve human goals with minimized action cost.

## 4.4 Training and Inference

**Training:** The training samples for the high-level planner consist of human instruction, current completed plans, current semantic feature maps and the groundtruth plan for the next step, and those for the low-level controller include plan for the next step, textual and semantic features for current border segments and the groundtruth action sequences representing primitives, location and targets. The details of input and output are provided in Appendix B.

We leverage GPT-4 and the ProcTHOR simulator to generate the large-scale dataset to train the LLaVA-based high-level planner. We annotate several seed instructions and leverage GPT-4 to generate more instructions and corresponding plans based on the object list for each scene in the ProcTHOR, where samples with logical errors are filtered with PDDL parameters (Shridhar et al., 2020a). We then implement the generated plans in ProcTHOR and collect the navigation trajectories, RGB-D images, object locations and robot poses as the training data. Finally, the generated samples are parsed into high-level planning samples and low-level action data. We follow the supervised fine-tuning paradigm in LLM for training the LLaVA model in high-level planner and low-level controller, where we mask out $p_i$ and $\{a_j^i\}_{j=1}^{\tau_i}$ in the $i_{th}$ step. In the training stage, we propose to construct counterfactual samples to motivate the inference ability of the foundation model on EIF. Specifically, we remove the target objects in the scene descriptions from the original samples and replace them with target objects that have similar other properties such as usage through an artificial mapping method. Diverse contexts are created for the foundation model fine-tuning to mitigate overfitting to fixed scene layouts, with the expectation that the foundation model generates suitable target objects through mining connections between human instructions and the scene objects at a deep level. For example, the fine-tuned high-level planner can adaptively select interactive cups, mugs, or bowls based on the scene information to satisfy the demand of drinking water.

**Inference:** The high-level planner generates the planning for the next step based on the current RGB-D image and scene information represented by the semantic map, and the low-level controller predicts the action primitives, target object and interaction position based on the generated step-wise plan. The semantic feature maps are updated when implementing the low-level action sequences. The high-level planner will generate the plans for the next step only when the current low-level action is successfully achieved. The detailed process is illustrated in Appendix C.

## 5 Experiments

### 5.1 Implementation Details

**Training configurations:** We employed the LLaVA-7B architecture with the Vincua-1.3-7B pre-training weights for the high-level planner and the low-level controller, which is finetuned with our generated data by the LoRA strategy. For the visual encoder, we sampled 32 visual embeddings from each frontier in the semantic feature maps up to 256 tokens as scene information representation. We generated 2k instructions with three subparts (1386 target-specific short, 333 target-specific long and 332 abstract instructions) for 2509 scenes in ProcTHOR, which results in 30k groundtruth plans for training the high-level planner. We implemented the plans in ProcTHOR with $A^*$ algorithm to collect the expert trajectory as the groundtruth for training low-level controller. Target-specific short and long instructions mean those containing objects to be interacted (e.g. Place the egg in the bowl) for task achievements, whose number of step plan is respectively lower than 15 and not. Abstract

Table 1: Comparison with different EIF methods across different instructions in the ProcTHOR simulator, where LLM-P* represents the LLM-P without performing re-planning.

| Method | Normal-scale | | | | | Large-scale | | | | |
|--------|------|-------|------|-------|------|------|-------|------|-------|------|
| | SR | PLWSR | GC | PLWGC | Path | SR | PLWSR | GC | PLWGC | Path |
| **Target-specific Short** | | | | | | | | | | |
| LLM-P* | 27.86 | 23.49 | 41.50 | 35.35 | 25.27 | 17.16 | 11.70 | 33.25 | 22.87 | 65.75 |
| LLM-P | 28.36 | 23.62 | 42.33 | 35.57 | 27.47 | 18.63 | 12.64 | 35.21 | 24.63 | 63.47 |
| FILM | 5.97 | 5.97 | 11.17 | 11.17 | 16.55 | 0.49 | 0.49 | 4.84 | 4.84 | 33.68 |
| Ours | 45.77 | 40.75 | 57.88 | 51.14 | 23.29 | 45.09 | 34.41 | 58.21 | 43.13 | 59.11 |
| **Target-specific Long** | | | | | | | | | | |
| LLM-P* | 5.97 | 5.14 | 18.91 | 17.26 | 60.56 | 1.52 | 0.82 | 15.28 | 13.05 | 78.03 |
| LLM-P | 5.97 | 4.80 | 19.65 | 17.30 | 64.89 | 1.52 | 1.01 | 16.04 | 14.17 | 64.14 |
| FILM | 0.00 | 0.00 | 4.14 | 4.14 | 79.17 | 0.00 | 0.00 | 6.26 | 6.26 | 70.14 |
| Ours | 13.43 | 12.44 | 27.11 | 24.67 | 62.21 | 19.70 | 17.34 | 35.61 | 31.08 | 78.99 |
| **Abstract** | | | | | | | | | | |
| LLM-P* | 1.32 | 0.92 | 15.68 | 12.57 | 38.69 | 6.16 | 2.83 | 16.92 | 11.21 | 70.92 |
| LLM-P | 3.95 | 2.33 | 16.78 | 12.45 | 36.27 | 6.16 | 3.58 | 18.15 | 12.42 | 67.20 |
| FILM | 0.00 | 0.00 | 4.87 | 4.87 | 33.23 | 0.00 | 0.00 | 8.02 | 8.02 | 49.45 |
| Ours | 10.53 | 8.09 | 24.23 | 19.68 | 35.90 | 9.59 | 5.74 | 21.30 | 15.01 | 61.54 |

instructions do not contain the interacted objects in the instructions (e.g. Make a simple lunch for me). We also generate 201, 67 and 152 data for each subpart as the test set. We utilized 8 NVIDIA 3090 GPUs to finetune the high-level planner and the low-level controller for an hour in the training stage. More details are provided in Appendix B.

**Metrics:** Following the ALFRED benchmark (Shridhar et al., 2020a), we use success rates (SR), goal condition success (GC), path length and their path-length-weighted (PLW) counterparts for evaluation. SR means the ratio of the cases where the agent completely achieve the human instructions, and GC measures the ratio of objects in the state of goal achievements. PLWSR and PLWGC calculate SR and GC weighted by the expert trajectory planning step number divided by the actual execution step number, which measures the trade-off between performance and efficiency.

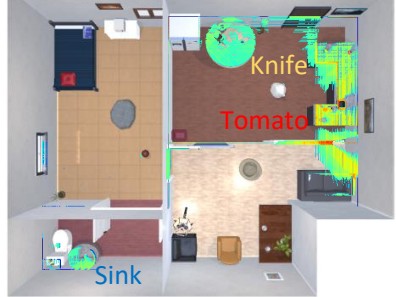

Figure 3: Example visualization of dynamic region attention weights.

**Simulated environments:** We perform extensive experiments in the ProcTHOR simulators, where the step size of translation and rotation for the agent is 0.25m and 90° respectively. ProcTHOR contains 10k house-level scenes with objects from 93 categories, where the agent receives $600 \times 600$ RGB-D images in the egocentric view. We divide the scenes into normal-scale ([0, 10]) and large-scale ([10, 16]) ones based on the side length of the room.

## 5.2 COMPARISON WITH BASELINES

Table 1 demonstrates the results on ProcTHOR for LLM-Planner, FILM and our method, where our approach significantly outperforms the state-of-the-art-method LLM-Planner. Although LLM-Planner utilizes the rich commonsense embedded in LLMs to generate plans for the agent, it fails to align the pre-trained LLMs with the scene information. The generated plans are usually infeasible due to the non-existence of the objects for interaction, and the re-planning module suffers from low success rate and low efficiency. On the contrary, our method construct the semantic feature maps which grounds the pre-trained multimodal LLMs to the realistic physical scene, and the unknown environment can be efficiently explored by understanding the visual clues for executable plan generation. In the target-specific short task setting, it is observed that our method outperforms LLM-Planner and FILM by 17.41% and 39.80% success rate in normal scale scenes, respectively. It is worth noting that our method loses less than 2% success rate in transferring to large-scale scenes, while LLM-Planner and FILM lose 34% and 91% success rate, respectively, which demonstrates the excellent scalability of our method in scene scales. Our approach remains leading in performance

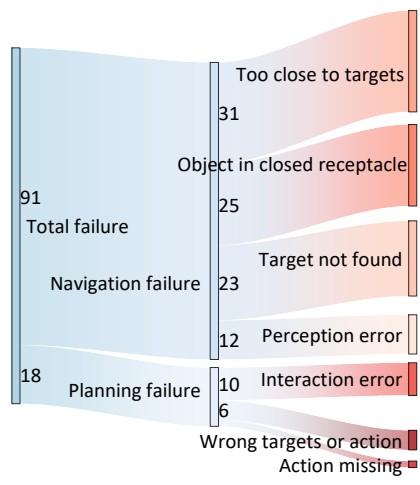

Figure 4: All failure cases on ProcTHOR simulator.

Table 2: Effectiveness of our generated plans and exploration actions.

| Method | GT | | Normal-scale & target-specific short | | | | |
|---|---|---|---|---|---|---|---|
| | Plan. | Exp. | SR | PLWSR | GC | PLWGC | Path(m) |
| Ours | ✓ | ✓ | 64.18 | 62.51 | 72.76 | 69.54 | 18.23 |
| | ✓ | - | 49.75 | 47.20 | 60.07 | 56.38 | 21.64 |
| | - | ✓ | 55.72 | 53.20 | 66.67 | 62.71 | 14.70 |
| | - | - | 45.77 | 40.75 | 57.88 | 51.14 | 23.29 |

Table 3: Ablation study of different scene feature maps.

| Method | Normal-scale & target-specific short | | | | |
|---|---|---|---|---|---|
| | SR | PLWSR | GC | PLWGC | Path(m) |
| No Map | 41.29 | 35.03 | 54.25 | 46.54 | 27.59 |
| No Attention | 44.78 | 39.02 | 56.63 | 49.40 | 24.54 |
| Random Attention | 44.27 | 38.24 | 56.72 | 47.96 | 25.90 |
| Ours | 45.77 | 40.75 | 57.88 | 51.14 | 23.29 |

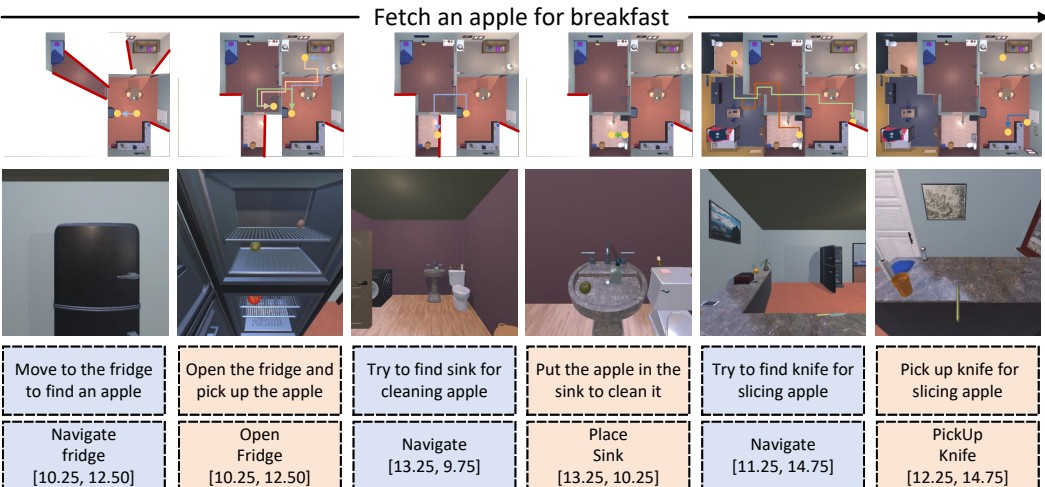

Figure 5: An example of EIF in unknown environments. The agent only navigates the task-related regions for visual clue collection with high efficiency, and generates feasible plans to complete the abstract instructions.

in more challenging target-specific long and abstract tasks. Meanwhile, the leading PLWSR and PLWGC metrics verify that our low-level controller can find the target object at a lower navigation cost. Moreover, the success rate of conventional methods (e.g., FILM) in the large-scale scenes is near zero, while our approach can achieve 9.59% success rate. Since the service robot is usually deployed in house-level scenes, our method is proven to be more practical.

We demonstrate the qualitative results in Figure 5, where we show the step-wise plan, the exploration process and the robot implementation during a whole sequence for EIF. In the beginning, the agent is initialized in the bedroom area and selects the navigation borders outside the room for exploration, as the instruction 'making breakfast' is irrelevant to bedrooms. During the navigation, the agent gradually knows to explore the kitchen area by observing the dining table and the counter, and it is even aware that opening the fridge may find food for breakfast due to the rich commonsense in our finetuned low-level controller. As a result, abstract instruction is achieved by serving diverse food for breakfast, where only related regions are navigated with high exploration efficiency in the unknown environment. Figure 4 illustrates the statistics of failure cases caused by different reasons. The failure mostly comes from unsuccessful navigation because of the large house-level scene, and the top reasons including 'too close to targets' and 'fail to see closed space' indicate that navigation algorithms should be designed with high compatibility of the subsequent manipulation.

Table 4: Ablation experimental results of exploration strategies in the task-specific short setting, where No Exp. and No Front. represent no exploration and no frontiers exploration, respectively.

| Method | Normal-scale | | | | | Large-scale | | | | |
|--------|------|-------|-------|-------|------|-------|-------|-------|-------|-------|
|        | SR   | PLWSR | GC    | PLWGC | Path | SR    | PLWSR | GC    | PLWGC | Path  |
| No Exp.   | 29.85 | 29.09 | 42.08 | 40.92 | 6.09  | 11.27 | 10.68 | 24.26 | 22.99 | 5.32  |
| No Front. | 41.29 | 35.03 | 54.25 | 46.54 | 27.59 | 36.76 | 26.91 | 49.35 | 35.51 | 52.38 |
| Ours      | 45.77 | 40.75 | 57.88 | 51.14 | 23.29 | 45.09 | 34.41 | 58.21 | 43.13 | 59.11 |

## 5.3 Ablation Studies

**Effectiveness of the high-level planner and the low-level controller:** We evaluated the variants of our method where the planner and the controller are respectively replaced with the groundtruth step-wise plans and groundtruth action sequences. It is important to note that some of the failure causes (e.g., too close to the target) illustrated in Figure 4 could not be resolved even with GT step-by-step planning and navigation goals. Table 2 demonstrates the results where the performance of our methods is close to that of the groundtruth, which indicates the effectiveness of our LLaVA-based planner and controller. Moreover, the performance of active exploration in low-level controller mainly influences the success rate, since it is important to find the correct objects to interact in unknown environments. Meanwhile, low-level controller significantly impacts the path length since directly exploring the related regions enables the agent to accomplish the instruction faster.

**Effectiveness of the online semantic feature map:** The semantic feature map provides visual information of explored regions for the planner and the controller to generate feasible plans and efficient actions, and we report the performance of different semantic maps to validate the effectiveness of our method. Table 3 demonstrates the results for the settings of no semantic maps, semantic maps with only category information, semantic feature maps without dynamic attention and our semantic feature maps. The results demonstrate that the implicit rich semantic features are necessary for effective exploration of unknown environments, and the dynamic attention also enhances the performance of the semantic feature map as it removes the information redundancy for the large house-level scenes. We also visualize the dynamic region attention when the agent builds the semantic feature map in the unknown environment as illustrated in Figure 3. For the instruction *Slice the tomato for salad*, the features of the kitchen area especially the tomato and the sink are considered with high attention(The green color represents greater weight), which indicates that the dynamic region attention learns relevant visual clues for feasible action generation.

**Effectiveness of active exploration:** Existing EIF frameworks often lack active exploration capabilities, making them difficult to deploy in unknown environments. Our approach addresses this limitation by utilizing pre-trained models to construct fine-grained semantic feature maps and leveraging foundation models to generate task planning and interaction actions based on these maps. Table 4 demonstrates the ablation experiments for different exploration strategies in the target-specific short setting. In house-level unknown environments, the no-exploration strategy reduces success rates by 15.92% and 33.82% for normal and large-scale settings, respectively, highlighting the importance of active exploration in unknown environment EIF tasks. The efficiency of active frontier exploration is demonstrated by the fact that the success rate of the navigation strategy without frontier exploration is reduced by 4.48% and 8.30%, respectively, with comparable navigation costs compared to our approach.

## 6 Conclusion

In this paper, we have proposed an EIF approach for unknown environments, where the agent is required to explore the environment efficiently to generate feasible action plans with existing objects to achieve human instructions. We first build a hierarchical EIF framework including a high-level planner and a low-level controller, and then build a semantic feature map with dynamic region attention to provide visual information for the planner and the controller. Extensive experiments demonstrate the effectiveness and efficiency of our framework in the house-level unknown environment. However, this work lacks real manipulation implementation and the designed navigation policy ignores the compatibility with manipulation. We will design mobile manipulation strategies for general tasks and implement the closed-loop system on real robots in the future.

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

## A    Extended Related Work

**Detailed comparison to FILM:**  Both our approach and FILM are hierarchical EIF frameworks containing high-level and low-level controllers. We provide the following detailed technical comparisons.

In terms of task planning, FILM employs language models (e.g., Bert) to classify task instructions into fixed categories (7 categories on ALFRED) and generates step-by-step planning based on fixed task parsing templates, which leads to poor scalability of FILM for complex task instructions. Instead, our approach uses a foundation model (LLaVA-7B) to parse task instructions based on the context, resulting in excellent scalability of our approach on complex task instructions (long sequences, abstract).

In terms of scene map construction, FILM constructs maps with explicit object categories, which results in the loss of significant fine-grained semantic information about objects (e.g., texture, usage). Instead, we construct scene maps with semantic features extracted from pre-trained models, fully exploring the semantic relationship between scenes and instructions, providing improved alignment of scene information with task planning.

In terms of reasoning, FILM dynamically samples subgoals based on environmental response and execution, and backtracks to previous subgoals to retry in case of interaction failure. Benefiting from the geometric and semantic information embedded in the scene semantic feature maps, our approach can dynamically generate high-level task planning and low-level interaction actions based on the environment state, as well as leverage the scene frontier to efficiently explore the unknown scene during reasoning.

**Detailed comparison to LLM-Planner:**  Both our approach and LLM-Planner are hierarchical EIF frameworks containing both high-level and low-level controllers, and utilize large language models for task planning. We provide the following detailed technical comparisons.

In terms of scene map construction, LLM-Planner constructs scene maps in the same way as FILM utilizing explicit semantics. In contrast, our approach employs semantic feature maps that can contain more fine-grained information.

In terms of reasoning, LLM-Planner directly utilizes the scene object category list as scene information, and the high-level controller generates the subgoals required to complete the instructions, and then invokes the previously working low-level controller to ground the subgoals to specific interaction actions. Meanwhile, LLM-Planner generates subgoals dynamically with re-planning mechanism to better adapt to scene changes. On the contrary, our approach directly uses latent semantic features as scene information, allowing the foundation model to fully exploit the relationship between scene objects and instructions to generate efficient task planning and interaction actions. Meanwhile, our approach can fully utilize the scene map geometry information to generate efficient exploration strategies compared to LLM-Planner to fully perceive the unknown scene information.

## B    Training and Testing Details

**High-level planner and low-level controller:**  In the supervised instruction fine-tuning stage, we reduce memory usage via DeepSpeed ZeRO-2. The learning rate for the feature mapping layer and the LLM backbone network is set to $2 \times 10^{-5}$, and the batch size is set to 8. Fine-tuning is performed for only one epoch. Since the semantic feature maps have been constructed through CLIP, the scene visual tokens are directly fed into the mapping layer without the visual coder during the training stage. The CE loss leveraged in the training process is represented by:

$$\mathcal{L} = -\mathbb{E}_{(\boldsymbol{X}_T, \boldsymbol{R}) \sim \mathcal{D}} \Big[ \sum_{m=1}^{M} \log p_{\boldsymbol{\theta}}(R_m | \boldsymbol{R}_{<m}, \boldsymbol{X}_V, \boldsymbol{X}_T) \Big] \tag{5}$$

where $\boldsymbol{X}_V$ denotes scene feature maps and $\boldsymbol{X}_T$ means input text prompt tokens. $\boldsymbol{R}_{<m}$ represents the output text tokens before the $m_{th}$ token $R_m$ and $M$ arenumber of output tokens. In this way, the pre-trained multimodal LLMs can be grounded to high-level planning and low-level control tasks in realistic scenes, where executable plans and actions are generated based on the scene representation.

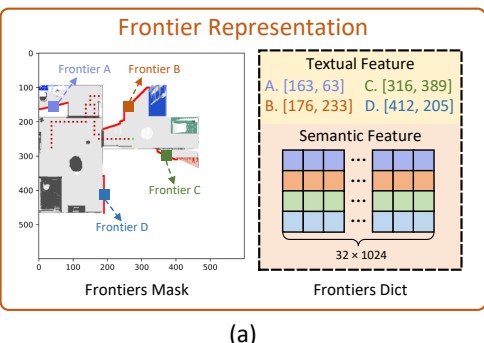 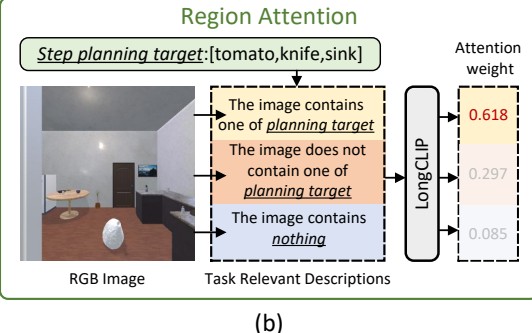

Figure 6: Details of frontier representation and region attention weights.

**Visual perception:** We selected 100k images from the captured expert trajectories as the training set for instance segmentation model Detic(Zhou et al., 2022), fine-tuning the pre-trained model with the learning rate of $1 \times 10^{-4}$ and performing 180k iterations. The batch size is set to 16 and the Adam optimiser is applied.

**FILM implementation details in ProcTHOR:** FILM contains three modules: task classifier, parameter classifier and instance segmentation. We use the generated instruction-following dataset to retrain the BERT-based task classifier and parameter classifier. Meanwhile, the instance perception module is replaced with the fine-tuned Detic from ProcTHOR scene to ensure a fair comparison. Depth information is directly used GT which is not the depth estimation model employed by FILM.

**LLM-Planner implementation details in ProcTHOR:** LLM-Planner mainly employs GPT-3 for task planning, since LLM-Planner is only partially open-sourced and the cost of invoking GPT-3's API is expensive, we employ the LLaMA-7B model instead of GPT-3. To further improve the performance of LLaMA-7B on EIF tasks, we fine-tune it using the generated instruction tuning dataset LLaMA-7B to ensure a fair comparison. The instance segmentation employs the same Detic, depth information from GT.

## C INFERENCE DETAILS

**Overview:** At the $i_{th}$ time, the agent surrounds to perceive the scene information and constructs semantic feature map $S_i$ and frontiers mask. Then, the agent will generate the $i_{th}$ high-level planning $p_i$ based on the scene information, user instruction $I$ and finished step-by-step planning $\{p_k\}_{k=1}^{i-1}$. The low-level controller generates specific interaction actions $a_j^i$, target objects $o_j^i$ and positions $l_j^i$ based on $p_i$, semantic feature $\{s_m^i\}$ and textual feature $f_m^i$: 1) If the $o_j^i$ is observed by the agent, $l_j^i$ will be the location recorded on the map; 2) If not be observed, $l_j^i$ will be the frontier position. During the inference process, each instruction $I$ performs up to 30 steps of high-level planning.

**Frontier representation:** We follow (Yu et al., 2023) to generate frontier masks that distinguish between known and unknown regions based on the occupancy

---

**Algorithm 1:** Inference Process

**input** : Human instruction $I$, high level planner $HP$, low-level controller $LP$, scene observation $\mathcal{O}$, maximum number of performing step $T$.

initialization: Random load into the unknown scene;

**for** $i \leftarrow 0$ **to** $T$ **do**
  Constructing semantic feature map $S_i$ via (3);
  Generate step planning $p_i$ based on $S_i$ via (1);
  **if** *end in* $p_i$ **then**
    | Break
  **end**
  Compute attention $w_i$ and update $S_i$ by (4);
  Generate action $\{a_j^i, l_j^i, o_j^i\}_j^{\tau_i}$ via (2);
  **for** $j \leftarrow 0$ **to** $\tau_i$ **do**
    | Execute $a_j^i$;
  **end**
**end**

---

map. Through connected component analysis, we obtain the mask of each frontier instance. We further remove frontiers with areas smaller than the threshold (150 pixels) to reduce redundancy exploration. We sample 32 visual embeddings as frontier tokens according to the frontier instance

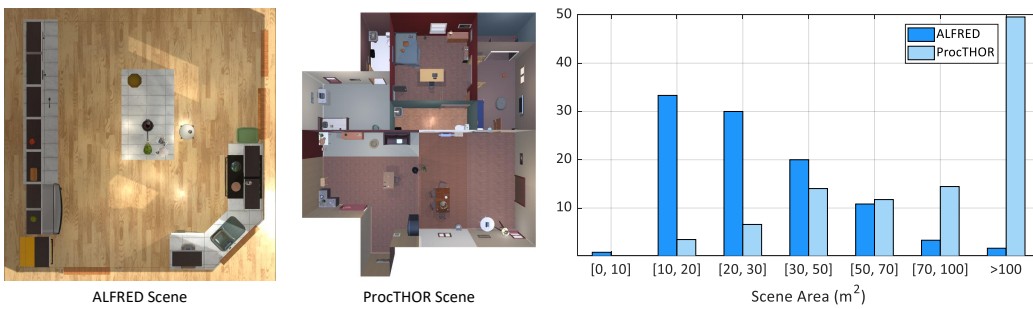

ALFRED Scene      ProcTHOR Scene

Figure 7: Comparison of ALFRED and ProcTHOR scene layouts and area distribution statistics results.

mask on the corresponding region of the feature map, while utilizing the coordinates of their centroids for the frontier text description. The specific representation is illustrated in Figure 6 (a).

**Region Attention:** Since the importance of the observed image for the completed instruction is different for each frame, it is desirable to assign higher weights to task-relevant visual embeddings to enable $LC$ to generate more efficient navigation exploration planning. $HP$ generates target objects that might be required to interact to complete instruction $I$ while gen-

Table 5: Results regarding frontier thresholds.

| Thresholds | SR | Path(m) |
|---|---|---|
| 70 | 45.77 | 36.50 |
| 100 | 46.27 | 30.41 |
| 150 | 45.77 | 23.29 |
| 200 | 43.28 | 19.32 |

erating $p_i$ and converts them into a sentence describing $L_{dec}$. However, measuring the relevance of an image to the instruction with only a single description is not discriminative enough to highlight task-relevant regions on the feature map. To this end, we add additional variants of descriptions to calibrate the relevance of images to their corresponding descriptions for exploiting the prior knowledge of the pre-trained models extensively. Specifically, we further expand $L_{dec}$ into $\overline{L_{dec}}$ and $L_{none}$ to match the input requirements of image and text alignment models such as CLIP. $\overline{L_{dec}}$ and $L_{none}$ describe the image as not containing the target objects and not containing the objects, respectively. We adopt LongCLIP (Zhang et al., 2024) to retrieve the similarity between $\left\{L_{dec}, \overline{L_{dec}}, L_{none}\right\}$ and RGB images as illustrated in Figure 6 (b), and consider the score of $L_{dec}$ as the attention weight.

# D  MORE RESULT

**Comparison of ALFRED and ProcTHOR scenes:** Figure 7 illustrates the scene layout and scene area statistics in ALFRED and ProcTHOR. The scenes in ALFRED are a single room (e.g., kitchen, bedroom), and agents deployed in ALFRED can easily perceive the complete scene information, which results in the agents being limited to generating plans in known environments. Meanwhile, the scene area in ALFRED is centrally distributed in `[10, 30]`, and previous approaches are less scalable in terms of scene size. On the contrary, the scenes in ProcTHOR are expansive house-level, and agents deployed in ProcTHOR can only perceive partial scene information, which requires the agents to construct real-time scene maps, in which feasible plans are generated with minimal exploration cost. The scene area in ProcTHOR is centrally distributed in `>100`, which is more scalable than ALFRED in terms of large-scale scenes.

**Influence w.r.t. navigation frontier construction:** Navigation frontier means the border between the known and unknown regions, which are represented by multiple segments. We only select the frontiers that are longer than a threshold as the candidates for agent navigation, because extremely short frontiers usually indicate corner regions that reveals uninformative information. Therefore, we can enhance the exploration efficiency significantly. Table 5 illustrates the success rate and path length for different thresholds. The results demonstrate that low thresholds result is redundant navigation with high path length, while high thresholds degrade the success rate because of important scene information. We set the frontier threshold to 150 pixels to achieve higher performance and navigation cost trade-off.

**Influence w.r.t. high level planner:** To further clarify the performance improvement of the model, we follow the FILM setting and use BERT to recognize the target objects from the instructions and generate high-level plans by filling the target objects into the corresponding parsing templates according to the predicted task categories. Table 6 illustrates the results demonstrating that changing the LLaVA-7B to BERT occurred with performance decreases, and the performance still outperforms the FILM due to the ability of the low-level controller to explore unknown regions to find the target objects.

**Influence w.r.t. foundation models:** Table 7 illustrates the results demonstrating that grounding the foundation model of e.g. GPT-4 to downstream EIF tasks using only prompt is

Table 6: Ablation experiment results for high-level task planner.

| Method | Normal & Short | | |
|---|---|---|---|
| | SR | GC | Path(m) |
| FILM | 5.97 | 11.17 | 16.55 |
| Ours w/ BERT | 24.38 | 39.81 | 20.64 |
| Ours | 45.77 | 57.88 | 23.29 |

Table 7: Ablation experiment results for the foundation model on the sub-test dataset.

| Method | Normal & Short | | |
|---|---|---|---|
| | SR | GC | Path(m) |
| GPT-4 | 35.00 | 53.33 | 19.39 |
| Conv-LLaVA | 40.00 | 58.33 | 17.40 |
| Ours | 45.00 | 61.67 | 21.03 |

not effective compared to fine-tuning MLLMs, which also suggests that the data synthesized by GPT-4 cannot be used directly for training and still requires post-processing. Meanwhile, the performance of different MLLMs (e.g. Conv-LLaVA (Ge et al., 2024)) does with little difference, consistent with the conclusion of the language model scaling law (Kaplan et al., 2020) that the main factor affecting language models of the same parameter size is the dataset scale.

**Qualitative results:** We demonstrate more unknown environment EIF execution sequences to reflect the superiority of our approach.

## E DATA

**Training Data:** Existing EIF datasets are still limited in instruction diversity and scene scale. We design a dataset synthesis framework to minimize the generation cost and increase the scale of EIF datasets, enabling agents to adapt to large-scale unknown scenes and complex tasks. Therefore, the dataset synthesis framework consists of two main stages. The first stage is to employ GPT-4 to generate extensive high-level planning with corresponding low-level actions based on prompt and scene information, then filter logical error samples with TextWorld (Shridhar et al., 2020b). The second stage is to execute the interactions specifically with the oracle in the simulator, grounding the generated plans and actions into the physical scene and collecting expert trajectories.

**TextWorld data generation:** We collect object lists contained in each scene as scene information in ProcTHOR, consisting of the location and size of each object. GPT-4 will generate task plans based on the object information and prompts. Specifically, we annotated 22 seed tasks manually to inspire GPT-4 to generate confirmed responses. Each response contains instructions, step-by-step high-level actions, and corresponding low-level actions. We further employ self-instruction (Wang et al., 2022) to ensure the diversity of instructions(The similarity filtering threshold is set to 0.9). Meanwhile, GPT-4 will generate PDDL parameters that satisfy the ALFRED benchmarks to verify the feasibility of the planning. The generated candidate samples are sent to TextWorld and check whether the task can be executed based on the PDDL parameters to ensure the quality of the training dataset.

**Grounding the generated plans:** The synthetic dataset that passes the PDDL check is fed into the ProcTHOR simulator for specific interactions. We collect navigation trajectories in ProcTHOR based on the planning generated in the first stage under oracle settings, which contain RGB images, depth maps, segmentation masks, and robot poses. According to the semantic feature map building approach presented in Section 4.3, we obtain real-time semantic feature maps $S_i$, frontier text features $\{f_m^i\}_m$, and semantic embedding $\{s_m^i\}_m$ sequences as the agents perform interactions. Based on the step-by-step planning generated by GPT-4, we split the above sequences into step-by-step instruction-following samples. As for $HP$, we feed instruction $I$, scene information $S_i$ and completed steps $\{p_k\}_k^{i-1}$ as prompts, expecting to generate the next step $p_{i+1}$ to be done. Meanwhile, in order to ensure the consistency of the high-level task planning, we require $HP$ to give the planning

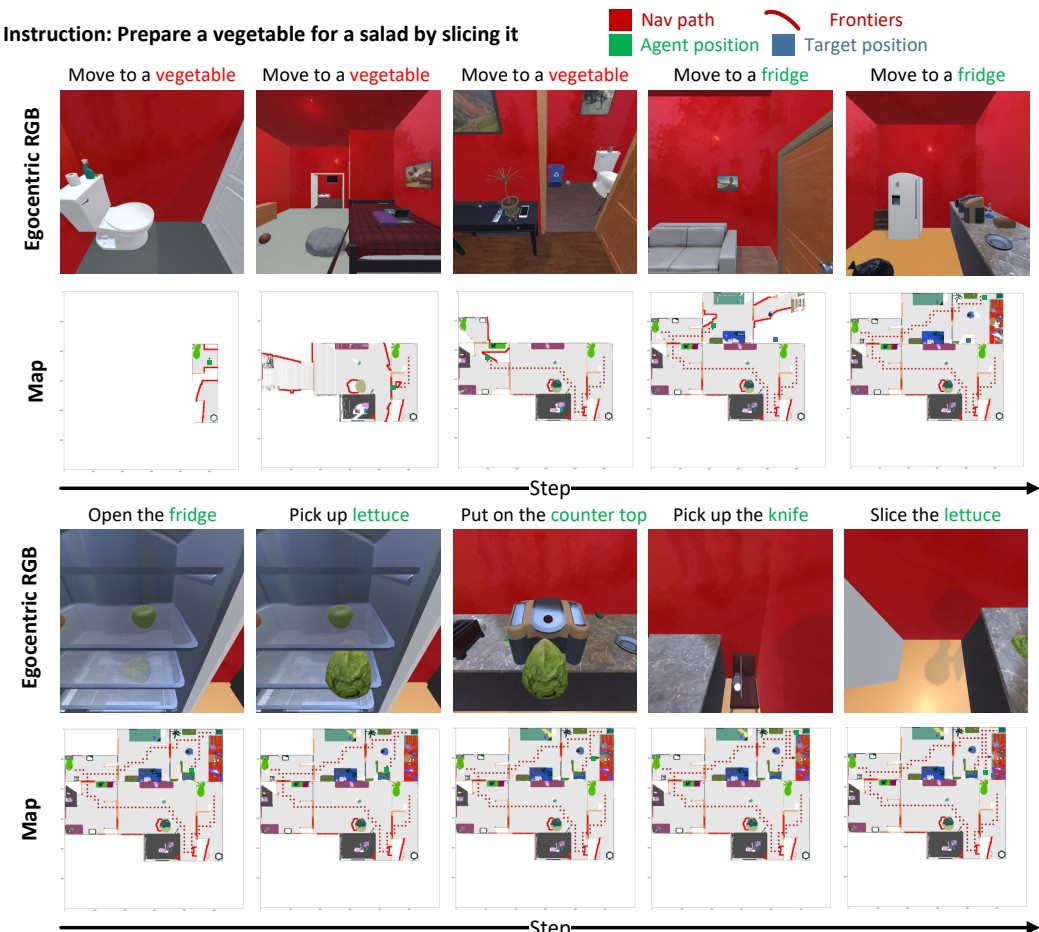

Figure 8: Our approach active search for lettuce in the fridge to complete the instruction.

of all subsequent actions as illustrated in Figure 10 (a). For $LC$, we take the current $p_i$, frontier text features $\{f_m^i\}_m$ and semantic features $\{s_m^i\}_m$ as prompt, and expect $LC$ to output the action primitives executed by the agent under the oracle setting. Specifically, if the agent observes the target object, LC generates the specific target and action primitive based on the input $p_i$. If it does not observe, we require $LC$ to generate the closest frontier to the oracle path as the next exploration region. $LC$ training samples are as illustrated in Figure 10 (b).

## F  PROMPT

Figure 11 briefly demonstrates the prompt words employed to inspire GPT-4 in generating the EIF dataset, which consists of the following four main parts:

**System Prompt:** Primarily designed to set up the GPT-4 contextual environment for generating task planning based on a virtual robot. Specifically, the system prompts contain the tasks that the GPT-4 needs to complete, and the role it needs to perform. Meanwhile, the rules that need to be followed for the response are also given in detail.

**Action Primitive:** It is applied to constrain the scope of the interaction action generated by the GPT-4 to ensure that it is executable. Each action primitive prompt contains both the action description and response format. Specifically, the action description provides GPT-4 with information about what each action primitive can perform in the simulator, e.g., Toggle to start an appliance, open to unlock containers, etc., while the response format informs GPT-4 about how the action primitive relates to the target object, e.g., the target of the Put action is a container rather than the object in the hand.

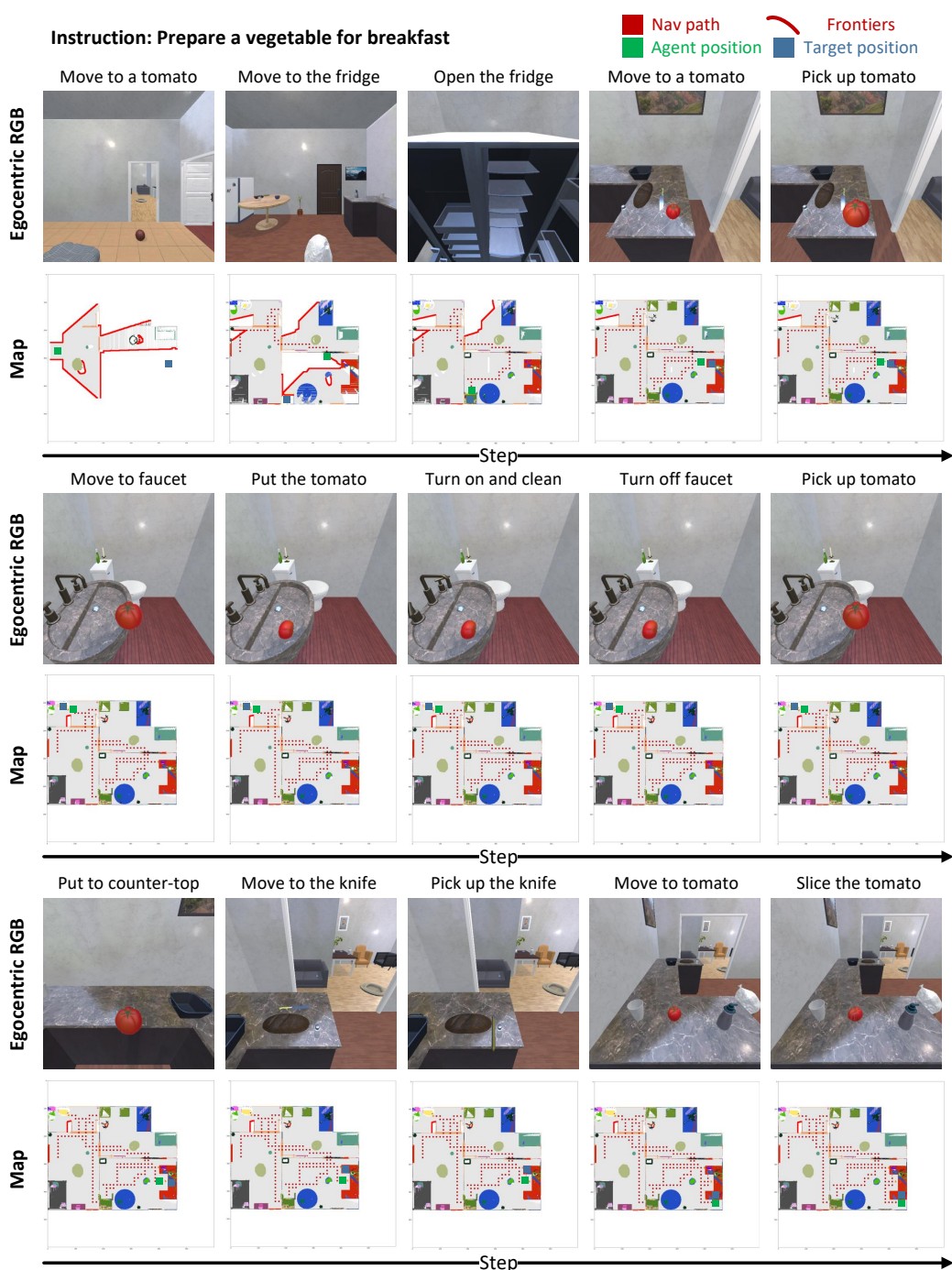

Figure 9: Our approach consistently completes complex instructions for preparing breakfast in large-scale unknown scenes.

**Response Format:** Ensure consistency in response format to parse specific instructions, planning and actions.

**PDDL Params:** Record the requirements for the completion of the instruction, including the target object and its state. An interaction is only successful if the state of the target object in the scene matches the PDDL parameter record.

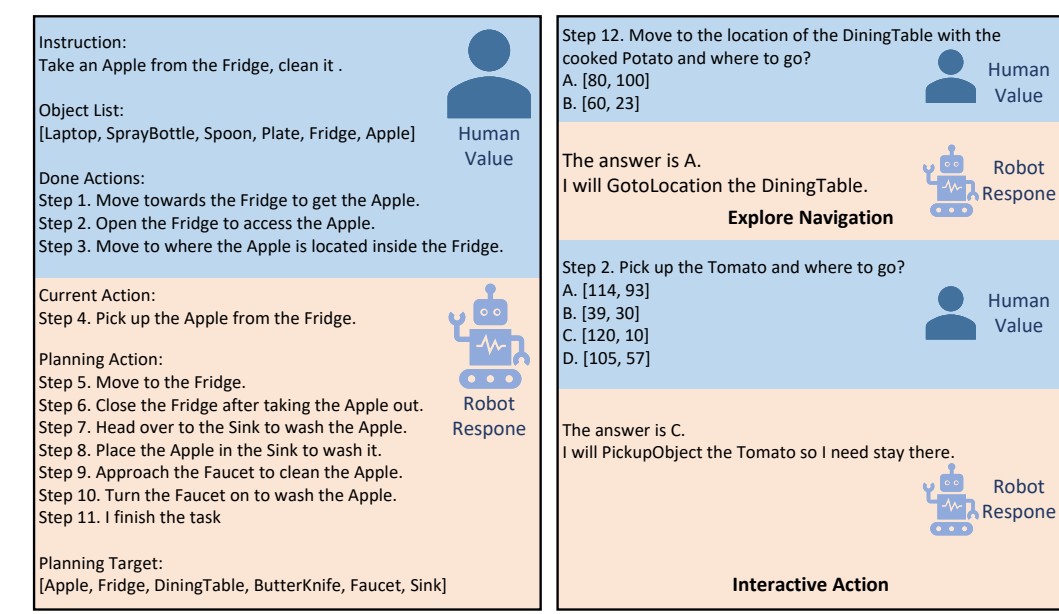

(a) High-level planner sample      (b) Low-level controller sample

Figure 10: Visualization of training samples for high-level planner and low-level controller.

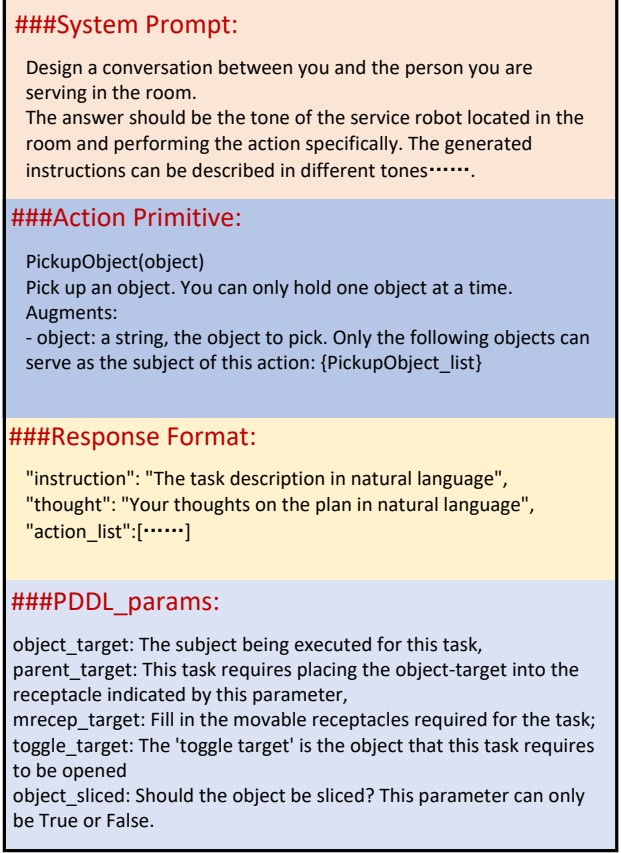

Figure 11: Prompt words for GPT-4 synthetic EIF dataset.