# OpenReview forum: "Embodied Instruction Following in Unknown Environments"
_ICLR.cc/2025/Conference — Submitted to ICLR 2025_

### Official Review · Reviewer_Vtnf · 2024-10-28

**Soundness:** 1
**Presentation:** 2
**Contribution:** 2
**Rating:** 3
**Confidence:** 3

**Summary:**

This paper introduces a modular approach for embodied instruction following tasks, which consist of three components: 1) high-level planning, 2) low-level control, and 3) building semantic feature maps. The proposed approach employs the LLaVA models and fine-tunes them on data generated by GPT-4 and the ProcTHOR simulator. Experiments show the comparison with baselines, ablation studies, and qualitative analysis to verify the effectiveness of the proposed approach.

**Strengths:**

Strengths
- The paper presents a new modular approach for embodied instruction following tasks, consisting of a high-level planner, low-level controller, and online semantic feature map.
- The paper conducts diverse analysis, including ablation studies and qualitative analysis, which helps to understand the proposed method.

**Weaknesses:**

Weaknesses
- I find no clear distinction between the “unknown environments” mentioned in this paper and existing test environments. Existing studies have also validated their proposed approach in unseen environments where embodied agents do not observe during training. Furthermore, prior work like LLM-Planner leverages the off-the-shelf object detector to feed the detected objects into LLMs, rather than providing all object categories with LLMs. How unknown environments are qualitatively or quantitatively different from unseen environments? Can authors verify the difference?

- In my current understanding, the paper conducts unfair comparisons. The proposed approach based on LLaVA is fine-tuned on 2k instructions generated by GPT-4, but the baseline model (LLM-Planner [1]) is a few-shot learning method that just uses 9 samples for in-context learning. The authors should compare their proposed method with more powerful models like EPO [2] which is trained on the in-domain data.


References

[1] Llm-planner: Few-shot grounded planning for embodied agents with large language models. Song et al. In Proceedings of ICCV, 2023.

[2] EPO: Hierarchical LLM Agents with Environment Preference Optimization. Zhao et al. In Proceedings of EMNLP, 2024.

**Questions:**

- In the paper, the train and test data share the same data distribution. Why do the authors use the word unknown environments?
- How the performance will be if the LLaVA model is trained on limited amounts of data (i.e., few-shot or zero-shot)?

---

> ### Author Response · Authors · 2024-11-24
>
> ## Q1：I find no clear distinction between the “unknown environments” mentioned in this paper and existing test environments.
> Unknown environment means that the agent can only perceive partial information about the scene and does not know what objects are in the scene.
>
> Unseen scene means that the agent has not seen the scene in the training set.
>
> Compared to ALFRED, the size of the unknown scene is much larger than the unseen scene.
>
> We demonstrate the differences in Appendix Figure 7.
>
> ## Q2: In my current understanding, the paper conducts unfair comparisons.
>
> The details of the LLM-Planner replication are specified in our Appendix B to ensure fair comparisons.
>
> ## Q3:  Why do the authors use the word unknown environments?
>
> As mentioned in Q1, unknown means that the agent does not know what objects are in the scene and has nothing to do with unseen
>
> ## Q4: How the performance will be if the LLaVA model is trained on limited amounts of data (i.e., few-shot or zero-shot)?
>
> Grounding the foundation model of e.g. GPT-4 to a downstream EIF task using only cue words is not effective compared to fine-tuning MLLMs.
> Meanwhile, the performance of different MLLMs does with little difference, consistent with the conclusion of the language model scaling law[1] that the main factor affecting language models of the same parameter size is the dataset scale.
>
> | Method            |   |        |     Small Specific   |        |          |
> |-------------------|-----------------|--------|--------|--------|----------|
> |                   | SR              | PLWSR  | GC     | PLWGC  | Path(m)  |
> | LLaVA-(zero-shot) | 0               | 0      | 0      | 0      | 0        |
> | GPT-4             | 35.00           | 31.41  | 53.33  | 48.02  | 19.39    |
> | Conv-LLaVA        | 40.00           | 36.41  | 58.33  | 53.95  | 17.40    |
> | Ours              | 45.00           | 39.56  | 61.67  | 54.22  | 21.03    |
>
> [1] Kaplan J, McCandlish S, Henighan T, et al. Scaling laws for neural language models[J]. arXiv preprint arXiv:2001.08361, 2020.

---

> > ### Comment · Reviewer_Vtnf · 2024-12-03
> > **Official Comment by Reviewer Vtnf**
> >
> > I would like to thank the authors for the response my concerns and questions.
> >
> > Similar to the comments by Reviewer VSiF, Q1 does not clearly differentiate the concept of "unknown environments" from the existing environments for embodied instruction following (e.g., ALFRED -- Shridhar et al. CVPR 2020). Moreover, Figure 7 in Appendix is not related to how unknown the environment in this paper is.
> >
> > For Q2, I have a follow-up question. The authors mentioned that they fine-tuned the LLaMA-7B model. How exactly the authors fine-tuned the model? More details should be discussed for reproducibility.

---

### Official Review · Reviewer_yycn · 2024-10-30

**Soundness:** 1
**Presentation:** 3
**Contribution:** 2
**Rating:** 5
**Confidence:** 4

**Summary:**

This paper proposes a novel embodied instruction following (EIF) method focused on scenarios that require executing complex tasks in unknown environments. The approach leverages a hierarchical EIF framework with a high-level task planner and a low-level exploration controller based on multimodal large language models. Utilizing semantic feature maps and dynamic region attention mechanisms, the method aligns task planning with the exploration process to fulfill complex human instructions. In extensive experiments within a large simulated environment, the approach demonstrated significantly improved success rates for complex tasks, such as preparing breakfast and organizing rooms, showing clear advancements over existing methods.

**Strengths:**

The use of scene feature maps to facilitate exploration in unknown environments, serving as visual input for VLMs while integrating textual instructions, presents an intriguing, well-founded, and promising approach. This framework combines planning and action seamlessly through the high-level task planner and low-level exploration controller, resulting in a cohesive and efficient system for embodied instruction following.

**Weaknesses:**

Experimental Environment Limitations: This method has been validated on a single simulator. It would strengthen the evaluation to include additional simulators, such as Habitat or iGibson, or to test the method on a real-world robotic platform to further demonstrate its robustness.
Limited Scalability and Computational Efficiency: The approach currently shows limited scalability and relatively low computational efficiency, which could impact its practical deployment in larger environments.
Lack of Comparative Models: The paper does not compare against certain state-of-the-art general models. For example, it would be interesting to see how a zero-shot baseline like GPT-4o performs across the various evaluation metrics.

**Questions:**

When will the code be open-sourced? Can the 7B version of LLaVA really handle such complex and diverse tasks that require precise outputs? Additionally, as far as I know, the 7B model of LLaVA has limitations in understanding depth maps. I’m surprised by the performance described in your paper!

---

> ### Author Response · Authors · 2024-11-24
>
> ## Q1: Lack of Comparative Models
>
> Thank you for your suggestions. For more foundation models, we show the results in Appendix Table 7.
>
> ## Q2: When will the code be open-sourced? Can the 7B version of LLaVA really handle such complex and diverse tasks?
>
> The code will be open source.
>
> The 7B foundation model has been utilized by several works on robotic task planning [1].
>
> In addition, there are also several works [2] that use the 7B foundation model to build robot generalists.
>
> Depth maps are only used to construct scene maps and are not sent as input to the VLM, we input visual features corresponding to the scene maps.
>
> [1] Kim M J, Pertsch K, Karamcheti S, et al. OpenVLA: An Open-Source Vision-Language-Action Model[J]. arXiv preprint arXiv:2406.09246, 2024.
>
> [2] Huang J, Yong S, Ma X, et al. An embodied generalist agent in 3d world[J]. arXiv preprint arXiv:2311.12871, 2023.

---

> > ### Comment · Reviewer_yycn · 2024-11-25
> >
> > The models in Table 7 of the appendix is not sota. The experimental simulation environment and the models used in this work are neither sufficient nor diverse enough. Therefore, I will maintain my current score.

---

### Official Review · Reviewer_BV4V · 2024-10-31

**Soundness:** 3
**Presentation:** 2
**Contribution:** 3
**Rating:** 5
**Confidence:** 5

**Summary:**

This paper introduces a hierarchical embodied instruction following (EIF) framework for autonomous systems to complete complex tasks in unknown environments. It addresses the limitation of conventional methods that fail to generate feasible plans when deployed in environments with unknown objects. The proposed solution includes a high-level task planner and a low-level exploration controller, informed by multimodal large language models (LLMs). A semantic representation map with dynamic region attention aligns task planning and scene exploration with human instructions. The framework efficiently explores unknown environments to discover relevant objects and generate feasible task plans. The authors report a 45.09% success rate in executing 204 complex instructions in large house-level scenes, demonstrating the effectiveness of their approach over existing methods.

**Strengths:**

1. The motivation of the paper is very good, taking into account situations in real environments where items may not be present at the time of planning.
2. The proposed framework is effective and also well-motivated. Combinations between high-level planners and low-level controllers are common, but the feature maps used in this work are interesting.
3. The paper is written in a fluid and well-organized manner.

**Weaknesses:**

1. In the main experiment the authors only compared LLM-P and FILM, the lack of other baselines weakens the superiority of the proposed approach. Here are three different types of baselines:

   (1) I would suggest adding some Object Navigation Method with LLM as Planner, such as MOPA[1], LGX[2].

   (2) Another very similar work, Demand-driven Navigation [3] also uses human instructions as input, has similar tasks, e.g., "I am thirsty" with "I need to drink water" and similar task motivation. I would suggest for baseline comparison or discussion at related work.

   (3) Multimodal LLM like GPT-4o, Gemini, LLaMA-3.2, Qwen2-VL can be directly used as baselines.

2. Need more details on task setting:

   (1) What is the step limit for each task?

   (2) How to determine task success and failure? If the agent completes the task with an undocumented solution (i.e., it was successful in the human viewpoint, but was not judged successful in simulator), how should the result be determined, or is there a high probability of this occurring?

   (3) If the task is partially successful, e.g. I want drink a cup of water, find the cup, but not the water, how should it be determined? Or conversely, would it make a difference if the water source is found but not the cup?

3. How much time and computational resources are needed for inference per task execution and how does it compare to baselines? Since the method uses a multimodal LLM like LLAVA, the speed of inference and computational resources are very important if it is to run in a real environment.

4. I believe the superiority of the methodology could have been enhanced if the authors could supplement experiments in real environments.

5. Need more details on the action space:

   (1) Does the agent need to have a knife in its hand to perform the Slice action? What happens if the agent does not have a knife in its hand but outputs a Slice? For other action, is there any precondition?

   (2) Can an agent only PickUp one item in hand at a time? Will it fail if it first PickUp item A and then item B?

   (3) How are Navigation low-level actions generated?

[1] MOPA: Modular Object Navigation with PointGoal Agents

[2] Can an Embodied Agent Find Your “Cat-shaped Mug”? LLM-Based Zero-Shot Object Navigation

[3] Find what you want: learning demand-conditioned object attribute space for demand-driven navigation

**Questions:**

see weaknesses.

---

> ### Author Response · Authors · 2024-11-24
>
> ## Q1:   The lack of other baselines weakens the superiority of the proposed approach
>
> Thank you for your suggestions. For more foundation models, we show the results in Appendix Table 7.
> Difficult to migrate across different simulators in a short time due to the various environments in which navigation work is deployed.
>
> ## Q2: Need more details on task setting
>
> (1) Step Limit: Maximum of 30 replanning steps per task and 10 trying steps per subtask.
>
> (2) Success Metrics: Determined based on the ALFRED benchmark, success for each task is determined by judging whether the target object conforms to a preset placement or state. Unrecorded solutions were considered failures. We believe this avoids unfair comparisons due to human preferences.
>
> (3) Partial Success: In the drinking water example, we judged success by two key steps. The first is whether the cup is in the sink; the second is whether the faucet is turned on. Completion of both is considered a successful instruction, while completion of one is considered a partial success.
>
> ## Q3: How much time and computational resources are needed for inference per task
>
> Thank you for your suggestion. We report the computational resources of the proposed method.
>
>
> |       Module      | High-level Planner | Low-level Controller | Perception |  Map  |
> |:-----------------:|:------------------:|:--------------------:|:----------:|:-----:|
> |     Memory(GB)    |       13.84        |        13.83         |   10.34    |  1.47 |
> | Execution time(s) |                    |         6.41         |            |       |
>
>
> |       Module      | High-level Planner | Low-level Controller | Perception | Map-ME |
> |:-----------------:|:------------------:|:--------------------:|:----------:|:------:|
> |     Memory(GB)    |       13.84       |        13.83         |   10.34    | 16.29  |
> | Execution time(s) |                    |         1.63         |            |        |
>
> ## Q4: Need more details on the action space
>
> (1) Execution is not required, but determining success is. If there is no knife in hand for slicing, the simulator will pass, but we are checking whether the agent has a knife in hand during the execution of the action. For the rest of the actions such as turning on the microwave oven, the microwave oven needs to be TURN OFF at this point.
>
> (2) Yes, there is only one object in the hand at a time.
>
> (3) The low level controller generates the boundaries for exploration and we use the A* algorithm to generate the cruise action.

---

> > ### Comment · Reviewer_BV4V · 2024-11-25
> >
> > I thank the author for their reply. After reading the rebuttal, I'm keeping my score.

---

> ### Comment · Reviewer_BV4V · 2024-11-25
>
> The reason about keeping my score is that:
>
> (1) I think object nav + LLM is a natural baseline. the former looks for objects, the latter plans and guides the former's exploration.
>
> (2) `Unrecorded solutions were considered failures`. Since task generation relies on LLM, some solutions may be missing, and the authors should compensate for this.
>
> (3) The computational resource consumption and inference time consumption are excessive, which diminishes the possibility of running in real-world deployments. I cautiously suggest that the authors experiment or discuss the possibility of real-world deployments.
>
> (4) If a domestic robot has been running around the house for a long time, locating the position of fixed and immobile objects like sinks and fridges seems to be an easy task (e.g. with YOLO detection or even human markers); so the environment is not really completely unknow.

---

### Official Review · Reviewer_VSiF · 2024-11-01

**Soundness:** 2
**Presentation:** 2
**Contribution:** 3
**Rating:** 3
**Confidence:** 4

**Summary:**

The paper proposes a hierarchical approach for embodied instruction following in unknown environments where room configurations may change due to human activities. The proposed approach consists of a high-level planner, a semantic feature map, and a low-level controller. Given an egocentric RGBD image, the agent updates its semantic feature map by weight-summing the newly obtained pixel-wise pretrained LongCLIP features and the old ones. The high-level planner receives as input a human instruction and the updated semantic feature map to generate high-level plans. For each high-level plan, the low-level controller predicts the target next location based on the semantic feature, text feature, and the high-level plan. The paper builds training and evaluation datasets based on ProcTHOR with GPT-4 and shows that the proposed approach achieves strong performance over the baselines.

**Strengths:**

- The paper is generally written well and easy to follow.
- The paper aims to address a challenging problem of EIF in environments that may change during the agent's task completion.
- The proposed approach achieves strong performance over the baselines.

**Weaknesses:**

- Unknown environments are not well defined. An environment is comprised of many elements, such as layouts, textures, object classes and instances, *etc.* It is not clear what is unseen to the agent. In addition, existing EIF benchmarks, like ALFRED, also evaluate agents in "unseen" setups. What is the difference between the unseen environments used here and the existing EIF benchmarks?

- Unlike prior EIF literature, the paper uses depth as additional input, but such depth information is unreliable in many cases. It is unclear if the proposed approach can be applied to RGB-only scenarios like prior EIF benchmarks and noisy-depth scenarios.

- The proposed approach maintains a semantic "feature" map as scene representation. Why not directly use the projected depth map and semantics (*e.g.*, object classes) associated with the corresponding pixels? The features in the map are updated in a weight-sum manner, but we can simply replace the part of the semantic map of what the agent is currently seeing with the old one.

- The low-level policy is learned to predict the target navigational location in a supervised manner. While simulated environments can provide such labels, obtaining them in real-world scenarios may not be available. Can this proposed approach applied to these scenarios as well?

- In Table 3, Ours vs. Random Attention and No Attention show marginal drops. I'm not sure if the proposed component is indeed effective.

- It looks like the proposed task setup closely resembles the ALFRED benchmark. Incorporating the proposed approach to more recent state-of-the-art methods here may imply the generalizability of the proposed setup.

**Questions:**

See weaknesses above.

---

> ### Author Response · Authors · 2024-11-24
>
> ## Q1: Unknown environments are not well defined
> Unknown environment means that the agent can only perceive partial information about the scene and does not know what objects are in the scene.
>
> Unseen scene means that the agent has not seen the scene in the training set.
>
> Compared to ALFRED, the size of the unknown scene is much larger than the unseen scene.
>
> We demonstrate the differences in Appendix Figure 7.
>
> ## Q2: Depth as additional input
> Here we need to clarify that depth information is only used to generate maps in this paper and previous EIF works, and has nothing to do with task planning or interaction action generation.
>
> Depth map information only affects the accuracy of the reconstructed scene map
>
> ## Q3: Why not directly use the projected depth map and semantics
> Visual features extracted with pre-trained models are more suitable for VLMs for task planning, and this has been applied in extensive work [1, 2]
>
> Direct explicit building semantic maps only records coarse semantic information, ignoring more fine-grained information such as texture, usage, etc.
>
> Simply replacing the old parts would lead to significant observation waste, which is not conducive to generating fine-grained scene representations. Therefore, we use weight summation to maintain the scene semantic feature maps.
>
> We supplement the experiments comparing semantic maps with semantic feature maps on partial testing sets.
>
> | Map Format           | SR     | Path  |
> |----------------------|--------|-------|
> | Semantic Map         | 36.36% | 16.46 |
> | Semantic Feature Map | 54.04% | 16.56	|
>
> ## Q4: Can this proposed approach applied to these scenarios as well?
>
> Navigation policies are obtained based on the correlation of frontier features with target objects, independent of real or sim scenes. We all use foundation models to construct scene representations and generation strategies with good zero-shot capability.
>
> ## Q5: It looks like the proposed task setup closely resembles the ALFRED benchmark.
>
> Both ours and AFLFRED are EIF tasks, with specific differences in the type of instructions and the scale of the environment.
> We show the differences in detail in the experimental section and Appendix B.
>
> [1] Huang J, Yong S, Ma X, et al. An embodied generalist agent in 3d world[J]. arXiv preprint arXiv:2311.12871, 2023.
>
> [2] Jatavallabhula K M, Kuwajerwala A, Gu Q, et al. Conceptfusion: Open-set multimodal 3d mapping[J]. arXiv preprint arXiv:2302.07241, 2023.

---

### Meta-Review · Area_Chair_pPJo · 2024-12-17

**Metareview:**

The paper received ratings that were all below the acceptance thresholds (5,5,3,3). The reviewers raised several concerns such as lack of a proper definition for unknown environments, lack of certain baselines, high cost of inference preventing real-world deployment, and unfair comparisons. The authors provided responses to the reviewers. While the rebuttal addressed some of the concerns, the reviewers still had concerns (details below). Therefore, the AC follows the recommendation of the reviewers and recommends rejection.

**Additional Comments On Reviewer Discussion:**

All reviewers responded to the author's rebuttal. Reviewer VSiF still has concerns regarding the definition of unknown environments, generalization to the real world, and the performance drop in Table 3. Reviewer BV4V still has concerns such as high inference time and missing solutions. Reviewer yycn is still concerned about the lack of sota baselines. Reviewer Vtnf is also concerned with the definition of unknown environments similar to Reviewer VSiF. Also, they need more details for reproducibility of the work. Overall, the rebuttal did not address the concerns.

---

### Decision · Program_Chairs · 2025-01-22

Reject